# Do Distance-Dependent Competition Indices Contribute to Improve Diameter and Total Height Tree Growth Prediction in Juvenile Cork Oak Plantations?

**Paulo Neves Firmino** [1,*], **Margarida Tomé** [2] and **Joana Amaral Paulo** [2]

1 Forest Research Centre, School of Agriculture, University of Lisbon, Tapada da Ajuda, 1349-017 Lisbon, Portugal

2 Forest Research Centre, Associated Laboratory TERRA, School of Agriculture, University of Lisbon, Tapada da Ajuda, 1349-017 Lisbon, Portugal; magatome@isa.ulisboa.pt (M.T.); joanaap@isa.ulisboa.pt (J.A.P.)

* Correspondence: pnfirmino@isa.ulisboa.pt; Tel.: +351-21-365-33-56

**Abstract:** Competition indices may improve tree growth modelling in high-density stands, found often in new cork oak plantations. Distance-dependent competition indices have hardly been considered for juvenile cork oak plantations since existing models were developed for low-density mature stands. This study aims at inspecting the potential of including distance-dependent competition indices into diameter at breast height (d) and total height (h) growth models for *Quercus suber* L., comparing several distance-dependent and distance-independent competition indices. Annual d and h growth were modelled with linear and non-linear growth functions, formulated as difference equations. Base models were initially fitted considering parameter estimates depending only on site index (S) and/or stand density (N). They were refitted, testing the significance of adding each competition index to the model parameters. Selected models included the best-performing distance-dependent or -independent competition indices as additional predictors. Best base d and h growth models showed a modelling efficiency (ef) of ef = 0.9833 and ef = 0.9900, respectively. Adding a distance-dependent competition index slightly improved growth models, to an ef = 0.9851 for d and ef = 0.9902 for h. Best distance-dependent competition indices slightly overperformed distance-independent ones in diameter growth models. Neither S nor N were included on best fitted models. If inter-tree competition is present in juvenile undebarked cork oak plantations, it does not yet strongly impact individual tree growth, which may diminish the importance of using, at this stage, more complex spatially explicit competition indices on predicting individual tree growth.

**Keywords:** *Quercus suber* L.; young plantations; growth models; Richards functions; Lundqvist–Korf functions; McDill–Amateis functions; difference equation; montado

## 1. Introduction

In Portugal, the establishment of new cork oak plantations began towards the end of the 1980s, installed under several and consecutive public support programs to reforest agricultural lands, also promoted in other European countries such as Spain and Italy [1]. Today these young plantations occupy an area of around 100,000 hectares in the country [2]. The management of these plantations is a concern for forest owners and presents additional challenges, since they differ considerably from the adult cork oak stands currently under production, characterized by low tree density values [3]. In fact, while the average stand density in the country is 78 trees ha$^{-1}$, and 80% of the stands have fewer than 120 trees ha$^{-1}$ [2], young plantations that are less than 30 years of age have an average stand density of 500 trees ha$^{-1}$ [4,5]. These high densities at younger ages, before the beginning of cork extraction, may imply the development of specific growth models and other management tools. Understanding and predicting cork oak development, under different tree density and spacing values, is therefore essential for forest management.

The present work develops models to predict tree diameter and total height growth, as the principal variables for the development of an individual tree model for young cork oak plantations.

The growth of individual trees can be influenced by the environment (site characteristics), tree age, along with stand density, inherent genetics or presence of inter-tree competition (e.g., [6]). Independently of the species and the environment, tree growth follows three distinct stages across the lifecycle, which can be represented by a sigmoidal-like function (e.g., [7]). Growth models based on sigmoidal functions with parameters dependent on environmental or stand conditions, allow prediction of tree development, information that is useful to adapt forest management to the local conditions. Most individual tree models have used either linear or non-linear functions [8], but growth functions are inherently non-linear, which is advantageous in terms of incorporating the fundaments of biological theory. Richards and Lundqvist non-linear growth functions [9,10], are examples that have been successfully applied to describe and predict cork oak diameter growth both in adult and young stands [7,11]. Total height growth of the same species was modelled by [12], using the McDill–Amateis function [13].

Most of the existing diameter growth models have been developed for cork oak trees that are already under cork production [7,12,14–16], usually designated as adult trees. Additional work was carried out focusing on undebarked trees, usually referred to as juvenile trees [11,17]. The distinction between these two stages is crucial for the cork oak species, especially for diameter growth since it is influenced by cork extraction [18]. The differences in cork growth trends between both phases require a distinct approach to diameter growth modelling, which makes the lack of models regarding this life period more evident.

The inclusion of competition metrics allows us to consider the impact of inter-tree competition on tree growth, thus allowing us to assess if tree growth is being affected by a lack of site resources, due to the share of water, light and/or nutrients with neighboring trees. This interaction is expressed by mathematical formulas, known as competition indices, which act as performance indicators of individual trees within a stand. Competition indices can be expressed as size variables, absolute or relative growth rates and density metrics, but are usually categorized as distance-dependent or distance-independent (e.g., [19]). Distance-independent indices are based only on tree and stand variables, but calculation of the distance-dependent ones imply knowing the exact position of each tree. Knowing tree positions allows us to incorporate inter-tree distances, consider zones of influence of each tree, or use local density metrics, allowing the formulation of more complex indices [19–21]. Some authors reject distance-dependent competition indices, associating them with the need to register tree co-ordinates in the field [19]. However, this is not necessary for applying distance-dependent individual-based tree growth models, as they may be provided by a stand structure simulator or, more recently, by remote sensing techniques. Information about spatial stand structure may also be simulated, by providing descriptive information about the stand level of aggregation, followed by the simulation of a stand with such a structure [19]. Assessing the potential of using these two competition index categories on the explanation of cork oak development would allow an evaluation of the benefits of using indices with the exact tree positions in growth predictions. It is important to recognize that, even if the improvement in growth prediction is not very large, using distance-dependent indices will allow the simulation of more realistic and complex thinnings.

The integration of competition indices in growth models has been widely used in forest studies (e.g., [6,22–24]). In the cork oak species case, some research is found on this topic [11,15–17], but modelling with distance-dependent indices has never been applied to juvenile trees growing in high-density plantations.

To clarify the need to use distance-dependent competition indices in cork oak growth modelling of young tree ages (previous to the first debarking), our work focused on: (1) modelling undebarked cork oak tree diameter and total height growth, using growth functions formulated as difference equations, with parameters depending on site index

and/or stand density; (2) assessing improvement on the best-performing models when expressing the parameters, depending on the value of a single competition index, testing a large set of distance-dependent and distance-independent competition indices.

## 2. Materials and Methods

### 2.1. Tree and Stand Measurements

A group of thirty-nine permanent rectangular plots, established in young cork oak plantations, was used to obtain individual juvenile (undebarked) cork oak individual tree data. This network of permanent plots has been established since 2007 by the ForChange research group (Centro de Estudos Florestais; Instituto Superior de Agronomia). The plots were installed inside pure cork oak even-aged plantations, covering a considerable part of cork oak species currently distributed in the Portuguese territory. The last measurement was carried out in 2020 under the scope of the present work.

Measurements were taken periodically, mostly every three years, with 20% of the data being measured in intervals of four, five and six years (12%, 2% and 6%, respectively). Stand age when measurements occurred ranged from 6 to 26 years, and stand density ranged from 80 to 877 tree ha$^{-1}$ (Table 1). Tree measurements included diameter at breast height (d), in cm, and total height (h), in m, collected for each tree in each measurement date. Tree crown width (cw), in m, was estimated based on the fixed effects model from [25]. Climatic information was characterized by minimum (Tmin), mean (Tmean), maximum (Tmax) of monthly temperatures and annual precipitation according to the 1981–2010 climatic normals, provided by the Instituto Português do Mar e a Atmosfera (IPMA), available at http://www.portaldoclima.pt/ (accessed on 30 July 2022). Further site description can be found on previous studies [17,25].

**Table 1.** Individual tree measurements (d and h) and stand variables descriptive statistics—Minimum (Min), Maximum (Max), Mean and Standard Deviation (SD)—calculated from the 39 permanent inventory plots, including the multiple consecutive measurements per cork oak.

|  | Min | Max | Mean | SD |
|---|---|---|---|---|
| d | 0.2 | 25.8 | 8.8 | 5.1 |
| h | 1.3 | 9.5 | 3.8 | 1.5 |
| t | 6.0 | 22.0 | 14.5 | 3.9 |
| N | 80 | 877 | 401 | 163 |
| dg | 0.3 | 24.3 | 9.6 | 9.6 |
| hdom | 1.7 | 9.8 | 5.3 | 1.7 |
| S | 13.5 | 18.6 | 16.7 | 1.2 |
| CC | 0.1 | 1.3 | 0.6 | 0.3 |
| Tmin | 8.5 | 13.2 | 9.8 | 1.3 |
| Tmean | 14.1 | 15.6 | 16.1 | 0.6 |
| Tmax | 18.3 | 21.4 | 22.5 | 1.5 |
| Precipitation | 493.3 | 943.7 | 652.3 | 131.1 |

d is the tree diameter at breast height (cm); h is the tree total height (m); t is the age of the cork oak plantation (years); N is the number of trees per ha; dg is the quadratic diameter (cm); hdom is the mean height of the 25 thickest trees per ha (m); S is the site index value (m); estimated with the model developed by [12]; CC is the sum of tree crown area divided by plot area; estimated from the diameter under cork; according to the fixed effects model from [25]; Tmin, Tmean and Tmax are the minimum, mean and maximum monthly temperatures observed in the group of used inventory plots; Precipitation is the annual precipitation observed in the group of used inventory plots.

The computation of distance-dependent competition indices implies the collection of individual tree co-ordinates. This information was collected in the field, and later confirmed using satellite images, taking advantage of the even-spaced positioning from trees inside the plantations. The measurement of permanent plots at multiple dates led to a collection of 7548 individual tree measurements. Trees that died between two measurements were not considered when calculating competition indices regarding the last measurement, as well as growth associated with field measurement errors. Trees were considered in the

plot border when situated on plantation lines closest to plot limits. These trees were used for competition indices calculation but excluded in the modelling phase. Additionally, for the development of the growth models, debarked trees at age t or t + i, trees with no measurable diameter at 1.30 m, and trees situated on plot borders were also removed, leading to a final dataset of 1673 diameter growth measurements and 2121 height growth measurements. Both data are shown in Figure 1.

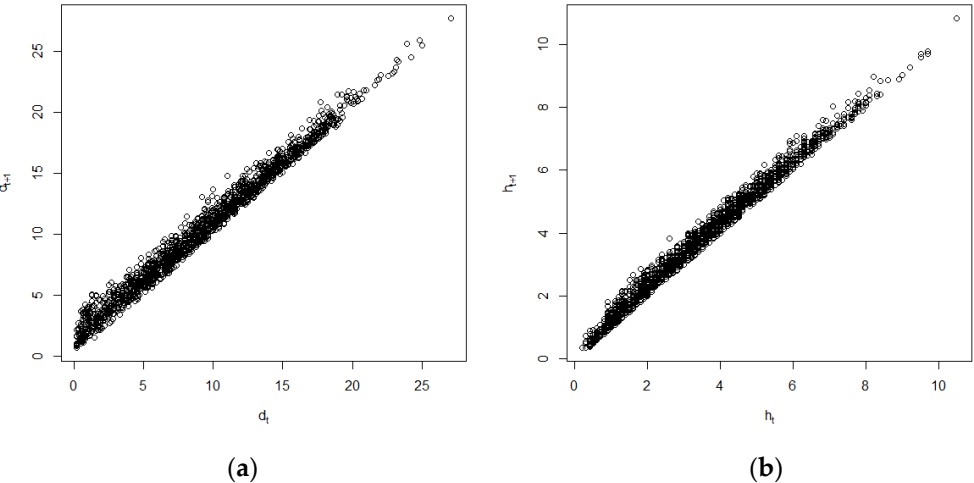

**(a)**                                    **(b)**

**Figure 1.** Plots of individual tree measurements: (**a**) diameter measurements, in cm, at year t ($d_t$) and at year t + 1 ($d_{t+1}$); (**b**) height growth measurements, in m, at year t ($h_t$) and at year t + 1 ($h_{t+1}$).

*2.2. Defining Neighbours and Computing Competition Indices*

An area of influence was considered for each individual tree, based on the expected size from the root system. Based on cork oak root system development studies [26,27], a circular area of influence was calculated with a radius equal to 2.5 times the average canopy radius of each tree, previously estimated from tree diameter (see Section 2.1). This area varied according to each individual measurement, and it was considered for the identification of neighbors: trees were considered neighbors when both areas of influence overlapped.

Tree competition was quantified using four stand density measures, a set of six distance-independent competition indices and seventeen distance-dependent competition indices. When calculated for both diameter or height growth, index abbreviation included ρ to express either d or h.

Four stand density indices were computed: number of trees per hectare (N), relative spacing (RS), spacing coefficient (SC) and percent crown cover (CC).

In the set of distance-independent indices, five were computed as relative dimensions of subject tree to the following stand variables: quadratic mean diameter ($R_{dg}$), maximum value of the variable ($R_{max}$), mean basal area ($R_{gm}$), dominant diameter ($R_{ddom}$) or dominant height ($R_{hdom}$), and a modified squared version of a mean measurement ($R_{mean}^2$). Additionally, we used plot level percent crown cover index, based on larger neighbours than the subject tree (CC > $\rho_i$).

We used ten distance-dependent competition indices based on the area of influence, previously tested on cork oak stands (see [28]). We also used six other indices previously applied to measure intraspecific competition in holm oaks in Spain (see [29]). Finally, we considered two modified formulations of widely used indices, tested on sessile oak in Turkey [30], integrating both diameter and total height dimensions on the same competition metric. Indices abbreviation, formulation and references can be found in Table 2.

**Table 2.** Mathematical formulas of the competition indices evaluated in this study.

| Abbreviation | Equation | Reference |
|:---:|:---:|:---:|
| *Stand density measures* | | |
| N | $\frac{10000}{A}n$ | |
| RS | $\frac{100}{h_{dom}\sqrt{N}}$ | [31] |
| SC | $\frac{100}{\overline{cw}\sqrt{N}}$ | [32] |
| CC | $\frac{\pi}{A}\sum_{j=1}^{n}cr_{ij}^{2}$ | |
| *Distance-independent competition indices* | | |
| $R_{gm}$ | $g_i/g_{mean}$ | [22] |
| $R\rho_{max}$ | $\rho_i/\rho_{max}$ | [33] |
| $R\rho_{dg}$ | $\rho_i/dg$ | [33] |
| $R\rho_{dom}$ | $\rho_i/\rho_{dom}$ | [33] |
| $R\rho_{mean}{}^2$ | $\rho_i{}^2/\rho_{mean}{}^2$ | [33] |
| $CC > \rho_i$ | $\frac{\pi}{A}\sum_{j=1}^{n}cr_{ij}^{2}\left(\rho_j>\rho_i\right)$ | [34] |
| *Distance-dependent competition indices* | | |
| nn | $\sum_{i=1}^{Nc}j_{(dist\geq dist_{ij})}$ | [35] |
| $N > \rho_i$ | $\sum_{i=1}^{Nc}j_{(dist\geq dist_{ij})(\rho_j>\rho_i)}$ | [35] |
| Sum$\rho$ | $\sum_{j=1}^{Nc}\rho_j$ | [36] |
| $\rho$ratio | $\frac{\rho_i}{\rho_i+\sum_{j=1}^{Nc}\rho_j}$ | [22] |
| $G > \rho_i$ | $\sum_{j=1}^{Nc}g_{(\rho_j>\rho_i)}$ | [35] |
| distnn | $Min(dist_{ij})$ | [35] |
| distnn $> \rho_i$ | nn $\rho_j > \rho_i$ | [32] |
| diffnn | $\rho_{nn}-\rho_i$ | [28] |
| Hegyi | $\sum_{j=1}^{Nc}\frac{\rho_j}{\rho_i}\frac{1}{dist_{ij}}$ | [37] |
| Hegyi.mod | $\sum_{j=1}^{Nc}\frac{d_j}{d_i}\frac{1}{dist_{ij}}\frac{h_j}{h_i}$ | [30] |
| dist$\rho$diff | $\sum_{j=1}^{Nc}\frac{\rho_j-\rho_i}{dist_{ij}}$ | [6] |
| MartinEk | $\sum_{j=1}^{Nc}\frac{\rho_j}{\rho_i}e^{\left[\frac{-16\times dist_{ij}}{\rho_i+\rho_j}\right]}$ | [38] |
| MartinEk.mod | $\sum_{j=1}^{Nc}\frac{d_j}{d_i}e^{\left[\frac{-16\times dist_{ij}}{d_i+d_j}\right]}\times\frac{h_j}{h_i}$ | [30] |
| Alemag | $\pi\sum_{j=1}^{Nc}\left(\left[\frac{\rho_i\times dist_{ij}}{\rho_i+\rho_j}\right]^2\left[\frac{\rho_j/dist_{ij}}{\sum_{i=1}^{Nc}\left\{\rho_j/dist_{ij}\right\}}\right]\right)$ | [39] |
| Lorimer | $\sum_{j=1}^{Nc}\frac{(\rho_j/\rho_i)}{(\sqrt{dist_{ij}/r})}$ | [40] |
| ClarkEvans | $\left(\sum_{i=1}^{n}\frac{min(dist_{ij})}{n}\right)\left(2\times\sqrt{N/A}\right)$ | [41] |
| NegExpSR | $\sum_{j=1}^{Nc}\left(\frac{\rho_j}{\rho_i}\right)\left[1/\exp\left(dist_{ij}+1\right)\right]$ | [38] |
| NegExpWSR | $\sum_{j=1}^{Nc}\left(\frac{\rho_j}{\rho_i}\right)\exp^{\left(\frac{-dist_{ij}+1}{\rho_i+\rho_j}\right)}$ | [38] |

where i is subject tree, j is a neighbour tree, d (cm) is tree diameter at 1.3 m, h is tree total height (m), $\rho$ is a tree size measure (d or h), dist is the radius of the circular area of influence corresponding to 2.5 the average canopy radius of each tree, $dist_{ij}$ is the distance between subject tree i and neighbor j, r is a defined fixed radius of 8 m centred on subject tree i, A is plot area (always 2000 m$^2$), cr is tree crown radius (m), (cw) is the mean of trees crown width, $\rho_{mean}$ is the mean of trees size measure, $\rho_{max}$ is the maximum of trees size measure, $\rho_{dom}$ is the mean of the dominant trees according to a size measure, N is stand density, n is the number of subject trees in the plot, Nc is the number of neighbors and nn is the nearest neighbor j to subject tree i.

### 2.3. Fitting Base Models and Models including Inter-Tree Competition

We developed individual tree diameter and total height growth models formulated as difference equations, considering an annual time interval. This implied estimating the annual tree growth of both modelled variables. Since most tree measurements were

triennial, we considered a linear growth in years between measurements. The tested models were selected from the literature, as growth functions previously used for cork oak modelling. Four models were considered: Linear [11], Lundqvist–Korf [7], Richards [11], and McDill–Amateis [12].

The fitting procedure accounted for two distinct phases. For the first phase, growth was modelled as a function of the respective base measurement at age t, site index and stand density [6,42,43], which were defined as base models. In the second phase, a single competition index was added to the base model with the best performance, fitting the competition models.

Difference equations can have multiple formulations per growth function according to the parameter (A, k, m) left as free [7,19,44]. From these combinations, we excluded from the analysis the Lundqvist–Korf function with m as free parameter, due to consistent problems in achieving convergence. McDill–Amateis growth function, when in differential form, is equivalent to the integral form of the Hossfeld IV function. It was developed to guarantee compatibility of dimensions and to consider the biological properties expected from a growth function [12,13,19]. The linear function was also tested, as the shape of d and h growth at this young stage seems to be in the linear phase of the growth function.

The asymptote parameter (A) estimates obtained were consistently low for an adequate biological meaning, leading to the need to fix diameter growth models asymptote to 200 cm [7]. The height growth models asymptote was selected as 20 m, according to the highest values of the height measurement collection of nearly 3500 adult cork oaks, monitored in the permanent plot network across Portugal of Forchange research group, from Centro de Estudos Florestais. Linear growth models tended to underestimate diameter growth values for trees with higher diameters, which led to, in this case, testing the additional inclusion of diameter at age t, as predictor to diameter linear growth models, to improve fitting. Thus, twenty-one difference equations were fitted to select the most suitable base model: variations of the Linear model (Linear-a; Linear-b), Lundqvist–Korf (LK), Richards (R) and McDill–Amateis (MA) functions (Table 3), according to the free parameter (-a; -b; -m; -k) combined with stand variables added as predictors (-N; -S; -N&S).

**Table 3.** Tested difference equations derived from the Linear function and the Lundqvist–Korf (L), Richards (R) and McDill–Amateis (MA) non-linear functions. In the designation of the functions, parameters -a, -b, -A, -k or -m represent the function free parameter.

| Function | Equation |
|---|---|
| Linear-a | $\rho_{t+1} = \rho_t - b(t - (t+1))$ |
| Linear-b | $\rho_{t+1} = a + (\rho_t - a)\frac{t+1}{t}$ |
| L-k | $\rho_{t+1} = A\left(\frac{\rho_t}{A}\right)^{\left(\frac{t}{t+1}\right)^m}$ |
| L-A | $\rho_{t+1} = \rho_t e^{k\left(\frac{1}{t^m} - \frac{1}{(t+1)^m}\right)}$ |
| R-k | $\rho_{t+1} = A\left(1 - \left(1 - \left(\frac{\rho_t}{A}\right)^{1-m}\right)^{\frac{t+1}{t}}\right)^{\frac{1}{1-m}}$ |
| R-A | $\rho_{t+1} = \rho_t\left(\frac{1-e^{-k(t+1)}}{1-e^{-kt}}\right)^{\frac{1}{1-m}}$ |
| R-m | $\rho_{t+1} = A^{\left(1 - \frac{\log(1-e^{-k(t+1)})}{\log(1-e^{-kt})}\right)} \rho_t^{\left(\frac{\log(1-e^{-k(t+1)})}{\log(1-e^{-kt})}\right)}$ |

where $\rho_t$ is the base tree diameter or height measurement (d or h) at age t, $\rho_{t+1}$ is the tree measurement at age t + 1, t is tree base age t, t + 1 is tree age one year later, a, b, k and m are model parameters and A is the asymptote parameter.

On the second phase, competition indices (CI) were added individually to the selected base model, testing adding them to each one of the functions parameters.

To assess the improvement from adding each competition index to the base model, Akaike information criterion (AIC) was calculated, and a relative change metric was calculated using the modelling efficiency (ef) [45], formulated as:

$$\Delta ef = \frac{(ef_C - ef_B)}{ef_B} \times 100$$

where $ef_B$ is the respective base model modelling efficiency without the competition index, and $ef_C$ is the modelling efficiency after inserting the competition index.

Model fitting, competition indices calculation and statistics analysis were executed with the R software version 3.6.0 [46], using the minpack.lm package for non-linear functions fitting [47], and the car package for model evaluation functions [48].

### 2.4. Model Evaluation

Models were evaluated according to the following considerations: goodness-of-fit, predictive ability, biological meaning of estimates, absence of multicollinearity and conformity to the assumptions of homoscedasticity and normality of residuals.

Modelling efficiency, the proportion of variation explained by the model (ef), was also calculated to evaluate fitting capacity. Bias and precision of the obtained estimates (predictive ability) were assessed with the analysis of the leave-one-out residuals (or jackknife residuals), usually designated by PRESS residuals, for being often used to calculate the predicted residual error sum of squares [49]. Bias was analyzed by calculating the mean of PRESS residuals, and precision by calculating the mean of the absolute value of PRESS residuals (aPRESS), along with the 5 and 95 percentiles of the PRESS residuals distribution [43]. To assess model performance according to tree size, six diameter classes ([0–2.5[; [2.5–7.5[; . . . ; [17.5–22.5[; [d > 22.5[ in mm) and ten height classes ([0–1[; [1–2[; . . . ; [8–9[; [h > 10[ in m) were created, to analyze the plots of each one of the above mentioned statistics against tree dimension classes.

The presence of multicollinearity was evaluated, since it may lead to unstable parameter estimation, as it inflates standard errors of some or all the regression coefficients. Multicollinearity is observed if the marginal contribution of any independent variable is influenced by other predictors present in the model. Collinearity between predictors was evaluated with the variance inflation factor (VIF): a VIF value is estimated for each predictor, where a value of one means absence of multicollinearity, and values exceeding 5 indicate potential problems [49,50], leading to discarding of the model.

Model homoscedasticity was analyzed by plotting studentized residuals against the predicted values. Normality of the residuals was checked by observing the quantile–quantile plot.

An evaluation was performed to verify if the best fitting diameter growth distance-dependent and distance-independent competition models brought an improvement over the existent diameter growth model fitted for juvenile cork oak stands, which will be designated P-Linear diameter growth model [11]. To compare these three models, we validated them using only the triennial measurements from our data, a total of 1288 observations, as this is the time step of the existing model. The mean of residuals and mean of absolute residuals were used as comparison statistics.

## 3. Results

### 3.1. Diameter Growth Modelling

All tested models performed well for modelling diameter growth, with linear difference equations performing better than non-linear ones. Best base models were obtained with the linear difference equations Linear-a, Linear-b and Linear-b-S, which were selected for the second modelling phase. Table 4 shows diameter growth base models which verified model assumptions; predictors were significant and had adequate biological meaning. Non-linear -S or -N variations which fulfilled these requirements are not shown, as they were outperformed by linear models.

**Table 4.** Model bias and precision statistics for the diameter growth base models and for the models fitted including a distance-dependent (DD) or distance-independent (DI) competition index.

| Model | Type | Mpress | Mapress | $P_5$ | $P_{95}$ | ef | AIC |
|---|---|---|---|---|---|---|---|
| | Linear-a | <0.001 | 0.524 | −0.912 | 1.230 | 0.9833 | 3350.1 |
| | Linear-b | −0.055 | 0.568 | −1.057 | 1.154 | 0.9807 | 3598.9 |
| | Linear-b-S | −0.072 | 0.558 | −1.008 | 1.165 | 0.9817 | 3506.2 |
| $d_{t+1}$ base model | L-A | 0.292 | 0.660 | −0.913 | 1.690 | 0.9718 | 4230.4 |
| | L-k | 0.152 | 0.635 | −0.934 | 1.633 | 0.9746 | 4053.5 |
| | R-A | 0.301 | 0.672 | −0.959 | 1.719 | 0.9707 | 4293.7 |
| | R-m | 0.203 | 0.668 | −0.919 | 1.773 | 0.9719 | 4224.1 |
| | R-k | 0.239 | 0.701 | −0.985 | 1.842 | 0.9689 | 4393.2 |
| | MA | 0.263 | 0.701 | −0.951 | 1.861 | 0.9687 | 4403.3 |
| $d_{t+1}$ DI model | Linear-a | <0.001 | 0.508 | −0.919 | 1.163 | 0.9844 | 3243.8 |
| | | $d_{t+1} = d_t - (1.6610 - 0.4654CC > d_i - 0.0395d_t)(t - (t+1))$ | | | | | |
| $d_{t+1}$ DD model | Linear-a | <0.001 | 0.494 | −0.884 | 1.103 | 0.9851 | 3164.5 |
| | | $d_{t+1} = d_t - (1.6154 - 0.0776N > d_i - 0.0198d_t)(t - (t+1))$ | | | | | |

where Mpress is the mean of PRESS residuals; Mapress is the mean of the absolute value of PRESS residuals; $P_5$ and $P_{95}$ are the 0.05 and 0.95 percentiles of PRESS residuals distribution; ef is the model efficiency computed with the PRESS residuals; dt is the base tree diameter measurement at age t; $d_{t+1}$ is the tree measurement at age t + 1; t is tree base age t; $CC > d_i$, is the crown cover area from trees with greater diameter than subject tree i, inside the 2000 m$^2$ plot; $N > d_i$ is the number of neighbors with greater diameter than subject tree i and overlapping areas of influence.

When adding a competition index, 42% of the distance-dependent (20 in 48 models) and 13% of the distance-independent competition models (9 in 39 models) verified all model assumptions, estimates associated with the competition indices were significantly different from zero and showed adequate biological meaning. Best fitted distance-dependent model included N > di in the Linear-a, showing a model efficiency of 0.9851, an improvement of Δef = 0.2% over the respective base model. Best fitted distance-independent model included CC > di also for the Linear-a model, with a model efficiency of 0.9844 and an improvement of Δef = 0.1% (Table 4).

Residual normality assumption was always fulfilled and no evidence of serious problems of heteroscedasticity were found (results not shown). The highest diameter class [d > 22.5[, with a lesser number of observations, showed most disparity on the PRESS residuals for any of fitted models (Figure 2).

Validation of the P-Linear diameter growth model showed less suitability in modelling diameter growth, when compared the diameter growth models developed in the present research, using the same data. The P-Linear diameter growth model showed mean residuals of 2.098 and mean absolute residuals of 2.257, while the $d_{t+1}$ DD model showed mean residuals of −0.048 and mean absolute residuals of 1.52, and $d_{t+1}$ DI model showed −0.320 and 1.712, respectively (Figure 3).

*3.2. Total Height Growth Modelling*

Tested difference equations successfully modelled total height growth, with slight differences between functions. Best base models were obtained with L-k (L-k, L-k-S, L-k-N) and R-m (R-m, R-m-N). Since model fitting statistics were similar, these five models were used in the second modelling phase.

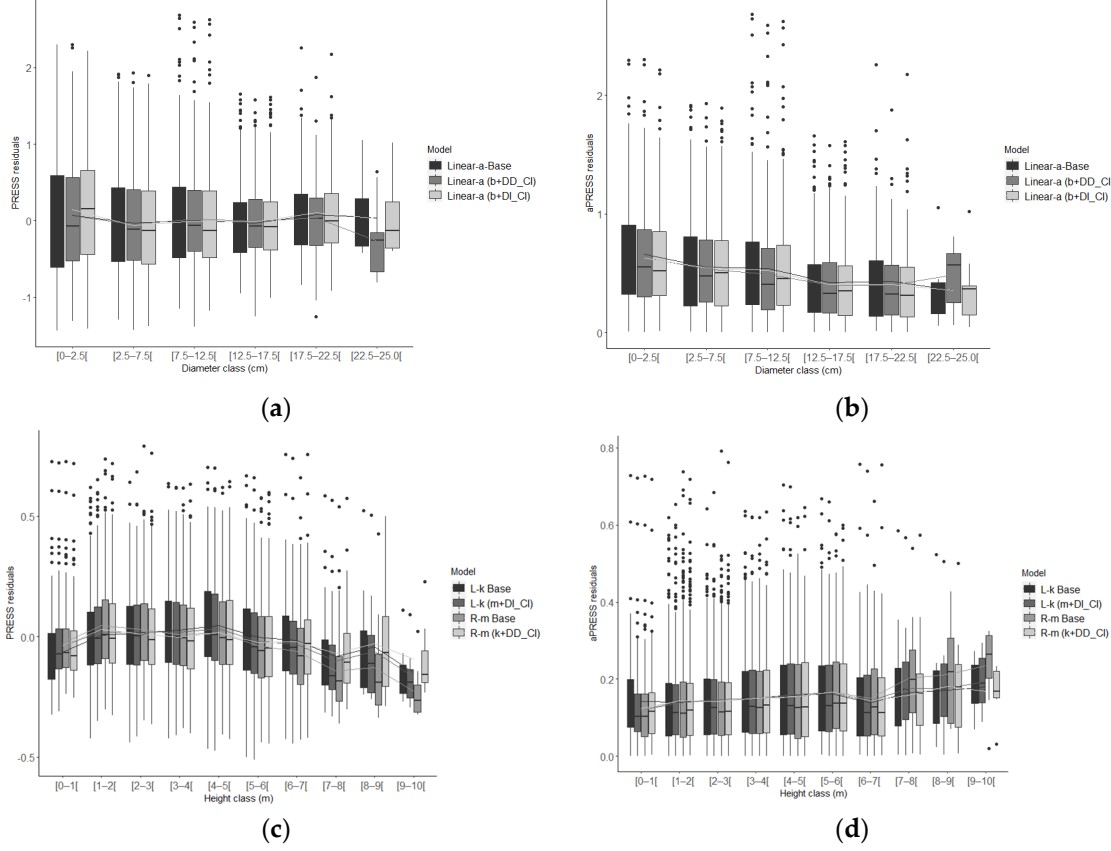

**Figure 2.** Boxplots and mean (lines) from diameter and total height growth PRESS (**a**,**c**) and aPRESS (**b**,**d**) residuals according to size classes: (**a**,**b**) prediction residuals statistics for the three diameter growth models: Linear-a base model (Linear-a Base); Linear-a when added the distance-dependent competition index N > $d_i$ to b parameter (Linear-a b + DD_CI) and Linear-a when added the distance-independent competition index CC > $d_i$ to b parameter (Linear-a b + DI_CI); (**c**,**d**) prediction residuals statistics for the four height growth models: L-k base model (L-k Base); L-k when added the distance-dependent competition index Sumh to the m parameter (L-k m + DD_CI); R-m base model (R-m Base) and R-m when added the distance-independent competition index RS to the parameter k (R-m-N k + DI_CI).

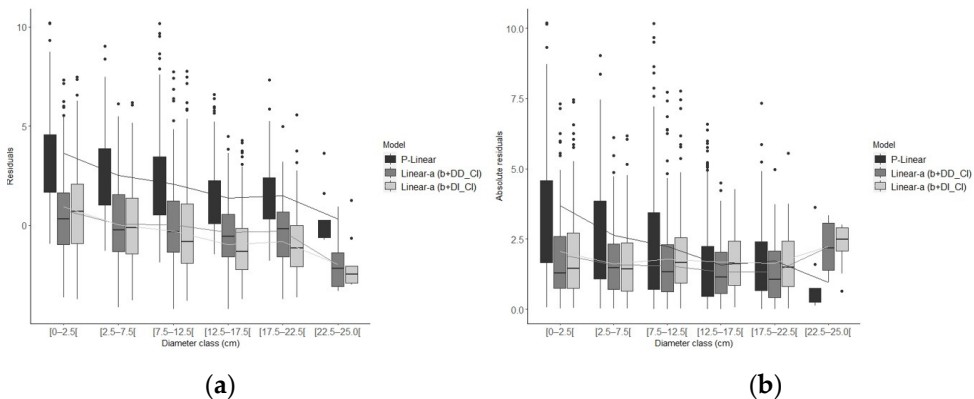

**Figure 3.** Boxplots and mean (lines) from diameter growth models validation according to size classes: (**a**) residuals and (**b**) absolute residuals from validating with three year growth data diameter the P-Linear diameter growth model (P-Linear), Linear-a when added the distance-dependent competition index N > $d_i$ to b parameter (Linear-a b + DD_CI) and Linear–a when added the distance-independent competition index CC > $d_i$ to b parameter (Linear-a b + DI_CI).

When adding competition indices, 9% of the distance-dependent (7 in 75 models) and 24% of the distance-independent competition models (13 in 55 models) improved when compared to the base model. The best distance-dependent model included Sumh in R-m, showing a model efficiency of 0.9902, improving 0.02% relative to the respective base model. The best distance-independent model included RS also in L-k, showing similar results. Table 5 shows height growth base models which verified model assumptions and predictors were significant/had adequate biological meaning. Base model variations of adding the site index to m parameter (-S) and/or adding stand density to parameter m or k (-N&S/-N) are only displayed for L-k, Richard-m and McDill–Amateis functions.

**Table 5.** Total height growth base models and models fitted while adding the best-performing distance-dependent (DD) and distance-independent (DI) competition index: model precision, residual statistics and model equations.

| Model | Type | Mpress | Mapress | $P_5$ | $P_{95}$ | ef | AIC |
|---|---|---|---|---|---|---|---|
| $h_{t+1}$ base model | Linear-a | <0.001 | 0.162 | −0.284 | 0.404 | 0.9885 | −719.3 |
| | Linear-b | 0.001 | 0.157 | −0.288 | 0.372 | 0.9890 | −816.9 |
| | L-A | 0.036 | 0.162 | −0.298 | 0.393 | 0.9879 | −602.3 |
| | L-k | 0.006 | 0.149 | −0.261 | 0.357 | 0.9900 | −1019.9 |
| | L-k-N | 0.006 | 0.147 | −0.265 | 0.348 | 0.9902 | −1052.7 |
| | L-k-S | 0.006 | 0.150 | −0.261 | 0.357 | 0.9900 | −1027.3 |
| | MA | 0.017 | 0.154 | −0.288 | 0.368 | 0.9892 | −845.7 |
| | MA-N&S | 0.012 | 0.153 | −0.287 | 0.369 | 0.9894 | −882.5 |
| | MA-S | 0.012 | 0.153 | −0.282 | 0.372 | 0.9893 | −878.2 |
| | R-A | 0.037 | 0.163 | −0.301 | 0.398 | 0.9877 | −577.2 |
| | R-m | 0.002 | 0.149 | −0.273 | 0.367 | 0.9900 | −1008.1 |
| | R-m-N | 0.002 | 0.149 | −0.277 | 0.366 | 0.9900 | −1010.3 |
| | R-k | 0.014 | 0.156 | −0.303 | 0.370 | 0.9889 | −779.5 |
| $h_{t+1}$ DI model | L-k | 0.004 | 0.148 | −0.262 | 0.357 | 0.9902 | −1049.3 |
| | | $h_{t+1} = 20 \left(\frac{h_t}{20}\right)^{\left(\frac{t}{t+1}\right)^{0.8792-0.0713RS}}$ | | | | | |
| $h_{t+1}$ DD model | R-m | <0.001 | 0.148 | −0.271 | 0.362 | 0.9902 | −1053.7 |
| | | $h_{t+1} = 20^{\left(1-\frac{\log(1-e^{-(0.0390-0.0002Sumh)(t+1)})}{\log(1-e^{-(0.0390-0.0002Sumh)t})}\right)} h_t^{\left(\frac{\log(1-e^{-(0.0390-0.0002Sumh)(t+1)})}{\log(1-e^{-(0.0390-0.0002Sumh)t})}\right)}$ | | | | | |

where Mpress is the mean of PRESS residuals; Mapress is the mean of the absolute of PRESS residuals; $P_5$ and $P_{95}$ are the 0.05 and 0.95 percentiles of PRESS residuals distribution; ef is the model efficiency computed with the PRESS residuals; $h_t$ is the base tree height measurement at age t; $h_{t+1}$ is the tree measurement at age t + 1; t is tree base age t; RS is the Relative Spacing index and Sumh is the sum of height values from neighbors with overlapping areas of influence to the subject tree i.

No evidence of violation of residual normality assumption or heteroscedasticity was found. PRESS residuals showed similar distribution across size classes and between fitted models, again except for the highest tree height class [9;10[ (Figure 2).

## 4. Discussion

This study successfully provided models to allow the prediction of diameter and height growth on juvenile and undebarked cork oak plantations. Our diameter growth model provides an alternative to the previous P-Linear diameter growth model, also focused on juvenile cork oak, using distance-independent competition indices. The proposed model is based on a considerably larger amount of data, collected within a wider stand productivity and plantation density ranges, and covers a considerable part of the cork oak distribution area in Portugal. Since the model was fitted with an annual time step, instead of the three-year time step considered in the P-Linear diameter growth model, it can be easily included in forest simulators that include the cork oak species, such as the SUBER v5.0 forest simulator [51], implemented in the sIMfLOR platform [52].

Total height growth was also successfully modelled, with R-m and L-k outperforming other tested difference equations. Non-linear functions were also tested for modelling total

height growth on dominant cork oak trees in Spain [12], with the McDill–Amateis function being used in a compromise between biological and statistical criteria. The same method was applied to *Quercus pyrenaica*, with the McDill–Amateis equation being the more suitable function for dominant height modelling [53]. Height growth models developed in this work provide an important tool for predicting cork oak height development in Portugal, particularly at the juvenile stage.

A difference between diameter and height growth models was visible, as linear models performed better for modelling diameter growth but not for height. It showed that cork oak diameter is in the linear growth phase at these ages, but this was not verified for height growth. A difference in the data may contribute to these results. Even though both measurements were from the same individual trees, diameter growth data excluded very young individuals which showed no measurable diameter at breast height, when they had still not reached 1.3 m of total height. These excluded youngest trees, the ones potentially at the exponential growth phase. To understand if this gap in age affected the modelling results or if diameter and height growth exhibit a distinct evolution in early age, it would require diameter measurements at the base of the tree.

Stand density and site index did not have an important role in the growth models developed, unlike the results from previous studies [6,12]. When either variable was added into a suitable model, the contribution was very low and simpler models excluding S or N were considered preferable. The range of site index values present in the data set was not very large (13.5–18.6), which may have influenced this result. Stand local density, characterized by distance-dependent competition indices, and the hierarchical position of the tree within the stand, characterized by distance-independent competition indices, performed better than stand density to explain individual tree growth, particularly for diameter growth. The low importance of stand density in the stands is in accordance with the findings in [17].

Adding competition indices slightly increased prediction capacity. This confirmed the weak presence of intraspecific competition in cork oak juvenile plantations [17]. Part of our data included very young plants, which leads to smaller tree size and areas of influence, thus lesser potential neighbor trees contributing to competition. The increase in individual tree size with simultaneous maintenance of high tree density will result in a gradual increase of intraspecific competition. To verify this assumption, maintaining monitoring on these stands is essential, as is more research to establish whether experimental thinning trials are crucial to clarify up to what age or tree size tree competition is not observed, and for those cases opening the possibility for finding a higher advantage of using distance-dependent indices.

The use of distance-dependent competition indices for modelling tree growth located in uneven-aged or mixed-species stands is another research topic interesting for cork oak forests or silvopastoral systems where cork oak is one of the primary species.

According to our data, neither the distance-dependent nor -independent competition indices were found to show an overall better performance. We identified CC > di (distance-independent) and N > di (distance-dependent) as the best-performing competition indices for diameter growth, with the distance-dependent index showing a slightly better performance. We identified RS (distance-independent) and Sumρ (distance-dependent) as the best-performing competition indices for height growth, both having similar performance. Various studies discuss which of these two classes of competition indices is more suitable for describing inter-tree competition, having distinct results depending on the species [54–56], regular or irregular stands [34,42], or age [29,43]. No specific competition index or group has been found to be generally superior, and their suitability differs depending on each forest context [19]. These conclusions were also obtained in our study.

## 5. Conclusions

Including distance-dependent competition indices slightly improved the fitting and prediction ability of individual tree growth models in young even-spaced plantations. The

effect of including these distance-dependent competition indices in uneven-spaced juvenile and adults cork oak stands should be analyzed. The improvement was not remarkable, reflecting the fact that plantations of the ages studied are not yet being strongly affected by inter-tree competition. Analyzing intraspecific competition on older cork oak plantations would provide important insights on the impact of competition with the progress of stand age. Results showed that cork oak juvenile stands are on the linear growth phase for diameter at breast height. Fitted models showed a very good predictive capacity of juvenile cork oak growth and are easy to run by users. Both fitted models and insights obtained from this work provide important tools to support forest management on juvenile plantations.

**Author Contributions:** Conceptualization and Methodology, P.N.F., M.T. and J.A.P.; Formal analysis, P.N.F. and J.A.P.; Writing—Original Draft Preparation, P.N.F.; Writing—Review & Editing, P.N.F., M.T. and J.A.P.; Funding Acquisition, P.N.F., M.T. and J.A.P. All authors have read and agreed to the published version of the manuscript.

**Funding:** This research was funded by the Forest Research Centre, a research unit funded by Fundação para a Ciência e a Tecnologia I.P. (FCT), Portugal (UIDB/00239/2020). This publication in Open Access was financed by the project UIDB/00239/2020 from the Forest Research Centre. First author was financed by FCT under the contract SFRH/BD/133598/2017. Third author was financed by FCT under the contract SFRH/BPD/96475/2013 and DL57/2016/CP1382/CT0027.

**Data Availability Statement:** The data presented in this study are available on request from the corresponding author. The data are not publicly available due to being part of an on-going network of permanent forest inventory plots.

**Acknowledgments:** We thank Diogo Castel-Branco, Sónia Faias, Tânia Oliveira, Valentine Aubard and Vanessa Inácio, among many others, who helped in acquiring field data along the various years.

**Conflicts of Interest:** The authors declare that they have no conflict of interest.

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
