# Peer review of "Do Distance-Dependent Competition Indices Contribute to Improve Diameter and Total Height Tree Growth Prediction in Juvenile Cork Oak Plantations?"

_forests, doi:10.3390/f14051066_

Round 1
Reviewer 1 Report
Review of “Do distance-dependent competition indices contribute to improve diameter and total height tree growth prediction in juvenile cork oak plantations?”
Interactions among neighboring trees (competition) are key determinants of forest growth (both primary and secondary growth). Correctly describing the relative influence of competition is critical to accurately predicting future plant productivity and plant community dynamics. This study explored the potential of including distance-dependent competition indices into diameter and total height growth models for Quercus suber L. by comparing several distance-dependent and distance-independent competition indices. They found adding a competition index slightly improved growth model.
Minor comments
1. Line 236. What “PRESS” mean? Please show the full name at the first time.
2. Line 240, Line 288 and Figure 2. “[0-2.5[; [2.5-7.5[; …; [17.5-22.5[; [d>22.5[ in 240 mm) and ten height classes ([0-1[; [1-2[; …; [8-9[; [h>10[ in m)…..” . Please revise [….[ to [……]
3. Line 255 and elsewhere. “….growth model fitted for juvenile cork oak stands by [11] was performed….” What [11] mean? Model? It looks like a citation.
Reviewer 2 Report
I like the paper. I have no neither objections, nor remarks.
Reviewer 3 Report
Dear Editor
The present manuscript investigated the importance of using distance-dependent competition indices in modeling the diameter and height of trees (Quercus suber L.) using linear and non-linear growth functions. These researchers reported that including the distance-dependent competition indices would improve individual growth models. The current manuscript is very attractive and its structure is completely suitable for publication in an international journal. However, some minor corrections below before final acceptance will improve the quality of this manuscript.
I am not a native-language person, but it seems that the text of the MS needs grammatical corrections.
The abstract should contain quantitative results.
Line 33: "around 100,000 hectares" instead of "around 100.000 hectares" (Use comma)
Line 93: "focused on" instead of "focused at"
Using "DBH" instead of "d" in expressing "diameter at breast height" is more common and it is better to do it throughout the MS text.
Geographical location (longitude and latitude) should be reported in the Materials and Methods section. Also, providing additional information about the studied area (i.e., environmental condition) will be useful for readers.
The results in Appendix A file should be presented in the text of the MS.
The conclusion should be developed and suggestions for future research should also be presented in it.
I am not a native-language person, but it seems that the text of the MS needs grammatical corrections.
